# The Impact of Workplace Violence on Headache and Sleep Problems in Nurses

**DOI:** 10.3390/ijerph192013423

**Published:** 2022-10-17

**Authors:** Nicola Magnavita, Luca Mele, Igor Meraglia, Marco Merella, Maria Eugenia Vacca, Anna Cerrina, Maddalena Gabriele, Marcella Labella, Maria Teresa Soro, Simona Ursino, Carmela Matera

**Affiliations:** 1Postgraduate School of Occupational Health, Università Cattolica del Sacro Cuore, 00168 Roma, Italy; 2Department of Woman, Child & Public Health Sciences, Fondazione A. Gemelli IRCCS, 00168 Roma, Italy; 3Local Healthcare Unit Roma 4, 00053 Civitavecchia, Italy

**Keywords:** workplace health promotion, healthcare workers, effort/reward imbalance, occupational risk, work ability, quality of care

## Abstract

Workplace violence (WV) is a significant occupational hazard for nurses. Previous studies have shown that WV has a reciprocal relationship with occupational stress. Headaches and sleep problems are early neuropsychological signs of distress. This cross-sectional study aims to ascertain the frequency of physical or verbal assaults on nurses and to study the association of WV with headaches and sleep problems. During their regular medical examination in the workplace, 550 nurses and nursing assistants (105 males, 19.1%; mean age 48.02 ± 9.98 years) were asked to fill in a standardized questionnaire containing the Violent Incident Form (VIF) concerning the episodes of violence experienced, the Headache Impact Test (HIT-6) regarding headaches, and the Pittsburgh Sleep Quality Inventory (PSQI) on sleep quality. Occupational stress was measured using the Effort/Reward Imbalance questionnaire (ERI). Physical and non-physical violence experienced in the previous year was reported by 7.5% and 17.5% of workers, respectively. In the univariate logistic regression models, the workers who experienced violence had an increased risk of headaches and sleep problems. After adjusting for sex, age, job type, and ERI, the relationship between physical violence and headaches remained significant (adjusted odds ratio aOR = 2.25; confidence interval CI95% = 1.11; 4.57). All forms of WV were significantly associated with poor sleep in a multivariate logistic regression model adjusted for sex, age, job type, and ERI (aOR = 2.35 CI95% = 1.44; 3.85). WV was also associated with the impact of headaches and with sleep quality. WV prevention may reduce the frequency of lasting psychoneurological symptoms, such as headaches and poor sleep quality, that interfere with the ability to work.

## 1. Introduction

Workplace violence (WV) is a common risk in nursing [1]. Although numerous studies have shown that it produces serious effects on the physical and mental health of the victims [2,3,4,5], interferes with their ability to work [6], and consequently reduces the quality of care, this phenomenon is severely underreported and poorly prevented. Globally, only a few countries include WV among the risks employers are obliged to consider and prevent [7].

The tendency to underreport the phenomenon should encourage a more proactive approach in the health surveillance of exposed workers: instead of waiting for spontaneous reporting, the physician should explicitly ask workers if they have experienced violence. Furthermore, since violence is known to cause stress, which can in turn predispose to violence [8,9], the doctor should look for symptoms that indicate the violence suffered before other and more serious pathological manifestations appear. Neuropsychological symptoms, such as headaches and sleep problems, could be potential indicators since they are often associated with violence. In fact, clinical studies have demonstrated that there is a relationship between violence and headaches. Patients with neurological disorders have often experienced abuse and violence [10]. Maltreated children [11], as well as abused women [12], may develop somatic and visceral central sensitivity syndromes, including tension-type headaches and migraine. Studies conducted in the workplace confirm the association between violence and headaches. Headache is among the most frequently reported negative effects of verbal and physical abuse on nurses [13,14]. Injustice in the workplace—a concept that also includes violence and bullying—is associated with headaches and sleeping problems in workers [15]. A case study on active workers showed that the subjects suffering from headaches were more likely to have been exposed to violence than the controls [16]. Sleeping problems are also consistently associated with WV. A meta-analysis revealed that workers exposed to violence have an increased pooled risk of developing sleep problems (OR = 2.55; 95% CI = 1.77–3.66) [17]. Another meta-analysis reported low to moderate evidence of the association between workplace bullying and sleep, with an estimated cumulative OR of 2.3 in cross-sectional studies and 1.6 in longitudinal studies for bullied workers compared to colleagues [18]. Headaches and sleep disturbances are also associated with each other. Disordered sleep, poor sleep quality, and insufficient or excessive sleep duration are known to trigger primary and secondary headaches [19]. Insomnia is an independent risk factor for headache chronicity [20]. An analysis of the factors influencing the onset of headaches in workers included disordered sleep among the predictive factors [21]. There is increasing evidence for the association of tension-type headaches with sleep disturbances, including insomnia, poor sleep quality, excessive daytime sleepiness, insufficient sleep, and shift working [22]. The co-occurrence of headaches and poor sleep and their mutual interaction could be due to a number of factors involving neurological circuits and psychological mechanisms still largely to be investigated. The fact that some neuronal sleep circuits are implicated in the pathogenesis of migraine suggests that neurological dysregulation may be at the origin of both headaches and disordered sleep [23]. Sleep problems are also associated with stress factors in the workplace, such as high work demands, job strain, bullying, and effort–reward imbalance, and could be prevented by social support at work, control, and organizational justice [24]. Shift work can also be associated with headaches [25] and sleep problems [26]. Stressors in a work environment can interact with each other and with individual characteristics (age and gender) to generate perceived occupational stress, which is subsequently related to the health status of the worker [27].

It is reasonable to expect that nurses who experience violence may develop headaches and sleep problems that might, in turn, reduce work capacity and interfere with the quality of care. To date there are no studies that have evaluated this possibility and verified the impact of these symptoms in the workplace. For this reason, we decided to ascertain the frequency of the exposure to WV in nurses and study the association of violence with headaches and sleep problems. The hypotheses examined were as follows:

**Hypothesis** **1** **(H1).**
*Workers who have suffered physical aggression, or a threat or harassment, in the year prior to the medical examination have a higher frequency of headaches and poor sleep quality than the controls who have not suffered violence.*


**Hypothesis** **2** **(H2).**
*Workers who have suffered violence have higher levels of effort/reward imbalance than the controls.*


**Hypothesis** **3** **(H3).**
*Exposure to violence and stress are associated with the impact of headaches.*


**Hypothesis** **4** **(H4).**
*Exposure to violence and stress are associated with sleep quality.*


## 2. Materials and Methods

### 2.1. Population

Secondary data were used from a workplace health promotion program conducted with the aid of a questionnaire during the medical surveillance of workers. In Italy and in other European countries, workers exposed to occupational hazards are required to undergo examination by an occupational physician to periodically monitor their health conditions. This study involved nurses and nursing assistants. Nursing assistants provide basic healthcare and support to patients in hospitals, correctional facilities, and in other local healthcare unit services, under the supervision of a registered nurse. Nurses are responsible for providing medical care to patients by monitoring vital signs, gathering information about healthcare problems, and offering emotional support. All the registered nurses and nursing assistants from a local healthcare unit who had been classified as “exposed to occupational hazards” for over a year and who had undergone their regular medical examination in the workplace in the period between 1 January 2021 and 31 December 2021 were invited to participate.

### 2.2. Questionnaire

The questionnaire contained, in addition to the personal data (sex and age), the Italian version of the Violent Incident Form (VIF), a validated questionnaire proposed by Arnetz for the registration of violent incidents in the healthcare workplace [28], which had previously been used in other Italian studies [3,4,5,6,29]. Exposure to the different forms of WV was investigated with binary-answer questions (no/yes), related to physical assault (physical assault means an attack, with or without weapons, that could cause or not cause physical damage), threat (a threat refers to the intention of causing physical damage), harassment (harassment is any annoying or unpleasant act—words, attitudes, actions—which creates a hostile work environment), and stalking (repeatedly following or harassing in circumstances that imply threat).

Headache was studied by a single no/yes answer (“Do you suffer from headaches?”), followed by an assessment of the impact of the headaches using the Headache Impact Test (HIT-6) [30]. HIT-6 contains 6 questions (e.g., In the past 4 weeks, how often have you felt too tired to do work or daily activities because of your headaches?), graded according to a 5-point Likert scale, from “never” to “always”. The 5 possible answers provide scores from 6 to 13; the result is a scale ranging from 36 to 78 points. The reliability of the scale, measured by the Cronbach’s alpha test, was 0.946 (excellent).

Sleep quality was measured with the Pittsburgh Sleep Quality Index (PSQI) [31], Italian version [32]. The questionnaire has 18 questions, which from 5 to 18 are graded using a 4-point Likert scale. The score resulting from the scale indicates reduced sleep quality; scores above 5 are an expression of poor sleep quality. The reliability (Cronbach’s alpha) of the questionnaire in this study was 0.891 (very good).

Occupational stress was measured using the Italian version [33] of the Siegrist Effort/Reward Imbalance model [34]. The questionnaire contains 10 questions, three for effort and seven for reward; the answers are graded according to a 4-point Likert scale. The weighted relationship between the two subscales, the Effort/Reward Imbalance index (ERI), is conventionally considered an expression of distress if higher than one. In this study, the reliability of the effort sub-scale was 0.854 (very good), and of the reward sub-scale, it was 0.641 (acceptable).

### 2.3. Ethics

The study was conducted according to the guidelines of the Declaration of Helsinki and approved by the University Catholic Ethics Committee (Project Number: 2896. Approval date: 5 December 2019). The participants signed the consent form, which was placed in their personal health files.

### 2.4. Statistics

#### 2.4.1. Sample Size

Sample size was calculated assuming a neutral distribution of the variables investigated (prevalence 50%), which is the condition that obliges the collection of the greatest number of responses. A very low margin of error (1%) and a 95% confidence level were also chosen for computing the minimum number of samples needed to offer the desired statistical power. We thus obtained the value of 523, i.e., 523 or more observations were needed to have a surveyed value that was within ±1% of the real value, with a confidence level of 95% [35].

#### 2.4.2. Tests

The distribution of the variables was initially studied using mean, median, and standard deviation. The association between the binary variables was investigated by means of the chi-square test for 2 × 2 tables; the comparisons between means were performed using the Student’s *t*-test for parametric data and the Mann–Whitney U-test for non-parametric quantities. The correlation between the continuous variables was studied using Spearman’s rho and Pearson’s r.

Using logistic regression, we calculated the risk of suffering from headaches or poor sleep for workers who had experienced workplace violence in the previous year. The risk was estimated as an odds ratio, with 95% confidence intervals. Following a hierarchical criterion, we initially considered the crude relationship between WV and headaches (Model I). The variables that were found to be significantly associated were introduced in the multivariate logistic regression model as confounding factors. We then adjusted the estimate by introducing age, sex, and category (Model II), as well as occupational stress, (Model III) as independent variables. Similarly, by means of hierarchical multivariate logistic regression models, we studied the relationship between WV and poor sleep.

To study how the variables that, according to the literature, could influence the productive impact of headaches or the extent of sleep problems, we devised multiple linear regression models in which violence, sex, age, job type, and stress were the independent variables, and the impact of headaches (measured with HIT-6) or, respectively, sleep quality (measured with PSQI) was the dependent variable.

The analyses were performed using IBM-SPSS (SPSS Statistics for Windows, Version 26.0. IBM Corp., Armonk, NY, USA).

## 3. Results

Most of the eligible workers (550 out of 552) agreed to participate. The number of participants was above the minimum number expected in the sample size calculation for a population of this size. The participants were predominantly professional nurses (55.2%), while the remainder were healthcare social workers and auxiliary nurses engaged in general nursing assistance (Table 1). Most of the workers were female (445, 80.9%), with an equal proportion in the two occupational groups. The age was on average 48.2 ± 9.98 years. The assistant nurses were significantly younger than the registered nurses (*p* < 0.001).

In the year prior to their routine examination, 41 workers (7.5%) had suffered at least one physical assault. The physical assaults, which had been carried out mainly by patients (86.5% of cases), but also by colleagues (13.5%), had affected the nurses and assistants with equal frequency. The most frequent type of aggression consisted of holding or jerking, pushing, or pulling hair; in less than a quarter of the cases, there had been scratching or pinching, spitting, and, more rarely, slapping or hitting, punching, kicking, or biting. The physical injuries had always been mild and never required admission to the emergency room. Less than a third of the incidents had been reported to colleagues and superiors and none had been reported as an accident at work. Of the 96 workers (17.5%) who reported threats, harassment, and non-physical assaults, a much greater frequency was observed in the registered nurses compared to the assistants (*p* < 0.001). Overall, 109 workers said they had been victims of some form of violence at work. The perpetrator was a patient in 73.4% of cases, a visitor in 7.3%, a colleague in 15.6%, and a stranger in 3.7%. The registered nurses had significantly greater exposure to different forms of WV than the assistant nurses (*p* < 0.001) (Table 1).

The assistant nurses had experienced significantly fewer attacks (26 vs. 83, Pearson’s chi-square 27.8, *p* < 0.001) than the registered nurses. They had lower ERI (0.79 ± 0.41 vs. 1.14 ± 0.52, *p* < 0.001), lower impact of headaches on working performance (41.7 ± 8.1 vs. 47.2 ± 11.4, *p* < 0.001), and better sleep quality (4.3 ± 2.9 vs. 6.8 ± 3.7, *p* < 0.001) than the registered nurses. Female sex was significantly associated with headache impact (45.5 ± 10.7 in female vs. 41.4 ± 8.4 in male workers, *p* < 0.001). Using point biserial correlation, we calculated the Spearman’s correlation coefficient between the continuous variables. Age was positively correlated with sleep problems and the effort/reward imbalance; ERI, headache impact, and sleep quality were significantly correlated with each other (Table 2).

The relationship between WV and neuropsychological problems (headaches and bad sleep) was investigated by logistic regression. Using univariate models, all the different forms of workplace violence were very significantly associated with headaches and bad sleep. The association between WV and bad sleep remained unchanged after adjustment for age, gender, type of work, and occupational stress. Physical violence was significantly associated with headaches even after correction for the above confounding factors, while the association between non-physical violence and headaches was no longer significant when the confounding factors were taken into account (Table 3).

Considering that physical violence has a significant impact on the presence of headaches, we investigated the association of physical violence with the occupational impact of headaches, measured using HIT-6 (Table 4). The physical assaults experienced in the year prior to the periodic medical examination were significantly associated with the impact of headaches. Headache impact was also increased by female sex, working as a registered nurse, and being exposed to high stress levels.

Experiencing any type of workplace violence in the year prior to the medical examination was a significant determinant of sleep quality (measured by the PSQI) in a multiple linear regression model (Table 5).

## 4. Discussion

Our study showed that experiencing physical abuse at work in the previous year (even if these episodes were apparently minor and generally unreported) was associated with a more than twofold risk of suffering from headaches and poor sleep. Violence was an independent risk factor for these symptoms. The introduction of corrections for gender, age, type of work, and work stress failed to modify the association between physical violence and the symptoms.

These results provided almost complete confirmation of the hypotheses formulated for this study. Only the relationship between non-physical violence and headaches was not significant after adjusting for the confounding variables, while all the other hypotheses were found to be valid. WV was associated with increased occupational stress, and the latter was a major determinant of headaches and sleep problems. The relationship between WV and sleep was stronger than that between WV and headaches.

Headache disorders, including migraine, tension-type headaches, and medication-overuse headaches, are the second leading cause of years lived with disability worldwide. They are associated with an impaired quality of life, a considerable loss of productivity, and high economic costs [36]. The WHO estimated that almost half of the adult population have had a headache at least once within the previous year [37]. Our field study confirmed that 48.8% of workers suffered from headaches, which were severe in over a third of cases; as a result, about one in five workers experienced a severe headache impact on their ability to work [21]. A Portuguese study estimated the average annual cost of each case of headache in employees to be over EUR 660 in direct costs. The indirect costs for absenteeism or presenteeism were even higher [38]. This economic burden may be more than tenfold greater in patients suffering from cluster headaches [39,40] or migraine [41]. The economic burden of sleep problems is substantial; in Australia, it is estimated that the financial costs of the most common sleep disorders are equivalent to 0.7% of the gross domestic product, and the non-financial costs are equal to 3.2% of the total annual burden of disease [42]. In the United States, in 2018, the direct costs of sleep disorders, which represent the smallest share of the total costs, amounted to USD 94.9 billion [43]. The association of the WV with headaches and sleep problems suggests that WV has a high and hidden cost, linked to its neuropsychological effects. The prevention of WV, in addition to being an obligation for the employer, may also be an economically advantageous choice for companies. The assessment of the economic viability of workplace violence prevention programs should take into account the economic impact of the associated psychoneurological disorders.

While the relationship between violence and sleep problems is consolidated in the literature, fewer studies have been conducted on its relationship with headaches. The Fifth Korean Working Conditions Survey demonstrated that all types of workers who had experienced WV were more likely to experience headaches and sleep-related problems [44]. Sleep problems are among the most common consequences of WV in Italian first aid and emergency staff [45], as well as in Chinese physicians [46] and nurses [2], Syrian doctors [47], and US homecare workers [48]. Moreover, the Swedish Longitudinal Occupational Survey of Health (SLOSH) suggested that exposure to WV may predict the development of sleep disturbances [49]. A recent meta-analysis of three previous studies [50] calculated the pooled relative risk of sleep disturbance in WV-exposed workers as being 1.22 (95% CI 1.09–1.37).

Few studies have been carried out on headaches among workers exposed to violence. The limited number of observations available refers mainly to nurses, who experience an increase in the frequency of headaches after exposure to WV [13,14]. In a systematic review, bullying, a form of violence exercised by colleagues (a high prevalence of which has been observed in healthcare settings), was associated with mental health problems, including psychological distress, and physical health problems, including insomnia and headaches [51]. Numerous stressors present in the workplace have been associated with the onset of headaches [52]. To support the existence of a relationship between WV and headaches, there are also a few clinical studies on traumatic antecedents in patients with migraine. The patient studies have shown that stressors experienced early in life are associated with migraine [53,54]. A history of abuse is often associated with greater migraine-related sensory hypersensitivity symptoms [55]. Migraine accounts for only a small fraction of the headache cases that occur in workers; however, these cases have a major impact on absenteeism and productivity [56].

Previous studies have already observed that exposure to violence varies by license type [57], and registered nurses are the most exposed to WV and have a greater risk of developing symptoms than assistant nurses [1,58]. In our observation, however, only non-physical assaults were much more frequent in registered nurses than in assistant nurses; this might be a consequence of a different awareness of what constitutes violent behavior. Studies showed that the most important reasons for not reporting WV were uncertainty regarding how and what types of violence to report [59] as well as the feeling that it was not important [60]. In point of fact, while physical attacks are easily identified by those who suffer them, other behaviors, such as ambiguous phrases that could constitute verbal threats, unwanted attention, or persistent, uncomfortable actions, interferences with own work ability, spreading rumors, talking in a negative way, giving undeserved criticism, or hanging around the workspace for no work-related reason, may not be readily recognized as WV by less-informed workers. Education and training have been shown to increase personal awareness of workplace aggression [61] and should therefore take precedence in providing workers with the knowledge and skills needed to prevent aggression and in preparing them to identify certain behaviors as violent. However, the difference in WV reporting between the nurse categories was lower than that observed between the departments of the same healthcare company, in which psychiatric and emergency services showed the higher aggression rates in previous studies [5,9]. Nurses’ resilience to WV is also likely to depend on numerous work-related factors; we have observed in previous studies that professional experience can be a protective factor [4], while the accumulation of unfavorable conditions, such as having to face an epidemic and therefore to fear for one’s health and that of loved ones [62], experiencing excessive workload, compassion fatigue, and lack of time for physical activity, relaxation and meditation [63], occupational and social isolation [64], and difficulties in assisting patients who refuse treatment and are aggressive towards staff [65], can foster a negative response that results in burnout or an intention to quit.

The effect of violence is mediated by numerous individual and occupational factors that affect not only the onset of symptoms, but also their severity and the impact they have on working capacity. Our studies confirmed the importance of gender, age, and above all occupational factors, job tasks, and work stress. These findings are in keeping with the literature. Headaches affect females more frequently than males [66]. Sleeping problems increase with age [67]. Occupational stress is a significant predictor of symptoms that include headaches and poor sleep [68,69].

This study has some limitations: one is due to the use of a convenience sample that prevents our results from being applicable to other occupational situations, even though there are no differences between the subjects studied in our survey and nurses from other healthcare companies. A further limitation concerns the WV retrospective data, which were merely anamnestic. The instrument used to measure the occurrence of violence in the year preceding the visit did not allow the measurement of the frequency and intensity of the events, nor did it allow knowledge of whether the workers had received treatment. However, it was difficult to render episodes of violence objectively on account of underreporting. Finally, the cross-sectional nature of the survey did not allow us to infer the causality of the symptoms. We cannot claim that the violence suffered in the previous year was the cause of the headaches and sleep disorders observed because some of these problems probably already existed. Nevertheless, the association observed makes us think that it is plausible that the violence exacerbated the intensity or recurrence of the symptoms. On the other hand, reverse causality, that is to say, suffering from headaches or sleep problems may have triggered the violence against the nurses, is an unlikely hypothesis that can be ruled out. Only longitudinal studies conducted in the workplace will be able to confirm the causal link between violence and neuropsychological effects.

This study lays the foundations for subsequent investigations into less obvious consequences of violence, such as neuropsychological effects, but it is above all an invitation to prevent WV. As it is not easy to prevent aggression, the implementation of a “zero tolerance” policy is important but does not go far enough. Education combined with training may have no effect on workplace aggression directed toward healthcare workers, but it is useful in increasing the knowledge of the phenomenon and encouraging positive attitudes between workers [61]. Organizational interventions that focus on the vector (patients), the victim (nursing staff), and the environment may result in a reduction in overall aggression, compared to practice as usual [70]. The best results could be obtained with the simultaneous implementation of educational, structural, and administrative interventions based on a participatory approach [71], although studies on the effectiveness of the various types of intervention have not yet provided definitive evidence [72].

## 5. Conclusions

In conclusion, this study demonstrates that violence experienced during work is associated not only with immediately visible consequences, such as physical injury, anger, fear, and other emotional phenomena, but also with neuropsychological problems such as headaches and sleep disorders that are highly insidious and may persist even several months after an attack. Headaches in the workplace have been associated with anxiety, depression, and metabolic disorders [21]. Sleep problems are significantly associated with immune and metabolic disorders, injuries, and traffic accidents [73]. Moreover, workers suffering from neurological symptoms such as headaches and sleep problems are unable to conduct their activities in the normal way. Consequently, violence against nurses may impair the quality of healthcare. This conclusion should prompt employers to implement effective measures for preventing WV. We strongly believe health managers are actively working towards this outcome.

## Figures and Tables

**Table 1 ijerph-19-13423-t001:** Characteristics of the population and comparison between registered nurses and nursing assistants.

	Registered Nurse (*n* = 302)	Nursing Assistant (*n* = 248)	Test	*p*
Sex, male (*n*, %)	58 (19.2)	47 (19.0)	Chi^2^	0.940
Female (*n*, %)	244 (80.8)	201 (81.0)		
Age (mean ± sd)	49.5 ± 8.6	46.3 ± 11.2	Student’s *t*	0.001
Physical aggression (*n*, %)	22 (7.3)	19 (7.7)	Chi^2^	0.867
Non-physical aggression	80 (26.5)	16 (6.5)	Chi^2^	0.001
All forms of WV	83 (27.5)	26 (10.5)	Chi^2^	0.001
Effort/reward imbalance (mean ± sd)	1.14 ± 0.52	0.79 ± 0.41	Mann–Whitney U	0.001
Headache (*n*, %)	171 (56.6)	96 (38.7)	Chi^2^	0.001
Poor sleep (*n*, %)	181 (59.9)	59 (24.7)	Chi^2^	0.001
Impact of headache (mean ± sd)	47.18 ± 11.42	41.71 ± 8.11	Mann–Whitney U	0.001
Sleep quality (mean ± sd)	6.79 ± 3.67	4.27 ± 2.94	Mann–Whitney U	0.001

**Table 2 ijerph-19-13423-t002:** Bivariate correlation between age, effort/reward imbalance, headache impact, and sleep quality. Values in the lower triangle are zero-order Pearson’s correlations; values in the upper triangle are Spearman’s correlations.

	Age	ERI	Headache Impact	Sleep Problems
Age	1	0.175 ***	0.045	0.145 ***
ERI	0.155 ***	1	0.226 ***	0.416 ***
Headache impact	0.081	0.278 ***	1	0.367 ***
Sleep problems	0.168 ***	0.393 ***	0.419 ***	1

Note: ***. Correlation is significant at the 0.001 level (2-tailed).

**Table 3 ijerph-19-13423-t003:** Association of workplace violence with headaches and poor sleep. Logistic regression analysis.

		Model I (OR, CI95%)	Model II	Model III
Physical violence	Headache	2.43 (1.23; 4.81) ***	2.54 (1.27; 5.10) ***	2.25 (1.11; 4.57) *
	Poor sleep	2.60 (1.33; 5.08) ***	3.08 (1.50; 6.31) ***	2.34 (1.11; 4.93) *
Non-physical violence	Headache	1.89 (1.20; 2.96) ***	1.51 (0.94; 2.41)	1.38 (0.86; 2.24)
	Poor sleep	3.63 (2.26; 5.84) ***	2.50 (1.52; 4.13) ***	2.09 (1.24; 3.52) ***
All forms	Headache	1.92 (1.25; 2.95) ***	1.62 (1.04; 2.53) *	1.48 (0.94; 2.33)
	Poor sleep	3.76 (2.39; 5.92) ***	2.92 (1.82; 4.70) ***	2.35 (1.44; 3.85) ***

Note: Model I: unadjusted; Model II: adjusted for sex, age, job; Model III: additionally adjusted for ERI; * *p* < 0.05; *** *p* < 0.001.

**Table 4 ijerph-19-13423-t004:** Association of physical violence with headaches. Multiple linear regression analysis.

Variable	Unstandardized B (CI95%)	Standardized Beta	*p*
Physical violence	4.90 (1.77; 8.04)	0.125	0.002
Female sex	4.08 (2.00; 6.15)	0.155	0.001
Age	−0.01 (−0.10; 0.07)	−0.01	0.768
Job, assistant	−4.03 (−5.79; −2.27)	−0.193	0.001
ERI	3.93 (2.18; 5.69)	0.192	0.001

**Table 5 ijerph-19-13423-t005:** Association of workplace violence with sleep quality. Multiple linear regression analysis.

Variable	Unstandardized B (CI95%)	Standardized Beta	*p*
Workplace violence	1.61 (0.93; 2.29)	0.182	0.001
Female sex	0.57 (−0.11; 1.24)	0.063	0.100
Age	0.03 (0.00; 0.06)	0.076	0.051
Job, assistant	−1.51 (−2.08; −0.94)	−0.211	0.001
ERI	1.86 (1.29; 2.43)	0.265	0.001

## Data Availability

The data will be deposited and available on the Zenodo repository.

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
