# Peer review of "The Impact of Workplace Violence on Headache and Sleep Problems in Nurses"

_ijerph, 2022, doi:10.3390/ijerph192013423_

Round 1

Reviewer 1 Report

Interesting article, with an extensive bibliography, and extensive argumentation. The presentation is adequate. However, the association-correlation between headache and sleep difficulties in nurses who have suffered violence in the workplace seems really obvious (explain); perhaps they can say that it is an exploratory investigation and that it raises many questions.

I think that the exciting thing would be to ask about the frequency and intensity of the violence suffered in that year and the treatments they have received at a psychomedical and psychosocial level, as well as adherence to medication and the effectiveness of therapeutic tasks. Many questions arise regarding the characteristics of the participating nurses: are they homogeneous? in addition to the detailed analysis according to gender, age, chronic diseases, and bio-psycho-social health.

It would be essential to reflect on all this in the text and to establish limitations in the discussion.

They must be more critical of the economic cost data they represent for companies. And if your study contributes something to this economic approach or not?

On the other hand, both in the introduction and in the discussion it would be opportune to make an orderly synthesis and avoid repetitions.

Author Response

Interesting article, with an extensive bibliography, and extensive argumentation. The presentation is adequate. However, the association-correlation between headache and sleep difficulties in nurses who have suffered violence in the workplace seems really obvious (explain); perhaps they can say that it is an exploratory investigation and that it raises many questions.

Response: The reviewer correctly pointed out a topic that we had already tried to highlight in the introduction: headaches and sleep disturbances are very interrelated. We have added a reference confirming this association. The association of headache and sleep disorders is obvious, but the association between violence and neuropsychological disorders is not. The aim of this study was to verify whether workplace violence is associated with these disorders.

I think that the exciting thing would be to ask about the frequency and intensity of the violence suffered in that year and the treatments they have received at a psychomedical and psychosocial level, as well as adherence to medication and the effectiveness of therapeutic tasks. Many questions arise regarding the characteristics of the participating nurses: are they homogeneous? in addition to the detailed analysis according to gender, age, chronic diseases, and bio-psycho-social health. It would be essential to reflect on all this in the text and to establish limitations in the discussion.

R.: The instrument used to measure the occurrence of violence in the year preceding the visit did not allow to measure the frequency and intensity of the events, nor to know if workers had received treatment and what adherence to treatment was. We have added this limitation in the discussion. However, the study was not intended to evaluate the effects of post-aggression treatment. This could be a useful tip for a later article. Nurses and assistant nurses who work in a healthcare company obviously perform different tasks depending on whether they are in an inpatient ward, surgical department, emergency room or psychiatric services and so on. The great variety of services also corresponds to a different level of risk of violence, as we have shown in a previous study conducted in the same company. We cited this study to clarify this point, but, in this research, we were only interested in verifying whether violence was associated with neuropsychic disorders. For this reason, we have considered the company's nurses and assistant nurses as a homogeneous group.

They must be more critical of the economic cost data they represent for companies. And if your study contributes something to this economic approach or not?

R.: We have reported data on the health costs of headaches and sleep disorders found in the literature. The purpose of this article was not to make a critical assessment of health economics assessments of these diseases. We have limited ourselves to mentioning that, if violence at work is associated with these problems, it can have an economic impact. To accommodate the reviewer's suggestion, we added the sentence: "The assessment of the economic viability of workplace violence prevention programs should take into account the economic impact of associated neuropsychic disorders"

On the other hand, both in the introduction and in the discussion it would be opportune to make an orderly synthesis and avoid repetitions.

R.: The reviewer did well to call for an orderly summary. Throughout the manuscript we have tried to be concise, without thereby being difficult to understand. Also in adding the comments that have been requested by the reviewers we have taken care to be very concise

Reviewer 2 Report

In this useful contribution to the literature on stress experienced by nurses, and the effects of such stress on certain aspects of health (headache and sleep disorder), the authors have used an opportunistic sample of 550 nurses who describe workplace violence (verbal and physical), and have  also completed validated measures of somatic symptoms. Statistical analyses are sound, and the conclusions are plausible.

What is missing - and this should be explained - is the type of nursing the informants were engaged in, such as ER work, COVID care, children's nursing, obstetric nursing etc. This might raise the question of why some persons, and not others, experience violence.

For this reviewer, who is engaged in longitudinal research with nurses in a European country, looking at severe stress, burnout, intention to leave nursing, depression and PTSD, these results are interesting: in our next data sweep I will include brief measures of somatic disorders, including headaches and sleep problems.

We have measured resilience in nurses, and have found that some nurses can cope with much stress without suffering psychological burnout etc; but others do not, and are likely to leave the profession. 

I would like the authors of this paper to add  to their discussion section (perhaps from line 338 onwards) a discussion, with additional references, of a variety of stressful factors faced by nurses, including the stresses of COVID care and associated patient mortality; and staff shortages (certainly in the  UK) because of these stressors, which may lead to downward spirals in patient care (e.g. long waits by patients for treatment and care leads to their aggressive behaviours). The authors must decide whether the somatic problems they measure may be related to more general malaise in nurses, and might predict burnout and intention to leave nursing.

In the acknowledgments, the (excellent) work of their English-language advisor is noted with the name:

Ms hank E.A. Wright

Is this a mistake - i.e. "hank" is a male name?

Author Response

In this useful contribution to the literature on stress experienced by nurses, and the effects of such stress on certain aspects of health (headache and sleep disorder), the authors have used an opportunistic sample of 550 nurses who describe workplace violence (verbal and physical), and have also completed validated measures of somatic symptoms. Statistical analyses are sound, and the conclusions are plausible.

Response: We are happy that the reviewer appreciated our work and took his / her time to try to improve our study.

What is missing - and this should be explained - is the type of nursing the informants were engaged in, such as ER work, COVID care, children's nursing, obstetric nursing etc. This might raise the question of why some persons, and not others, experience violence.

R.: We thank the reviewer for pointing out an important aspect, which we have considered in previous work conducted in the same company, in order to clarify the difference in the risk of violence in the various departments. In this study we only wanted to study the relationship between violence and psychoneurological disorders, so we considered the nursing staff as a whole. However, adhering to the reviewer's indication, we recalled in the discussion that the rates of violence are very variable according to the type of work carried out and we have cited two previous articles in which we have dealt with these variations.

For this reviewer, who is engaged in longitudinal research with nurses in a European country, looking at severe stress, burnout, intention to leave nursing, depression and PTSD, these results are interesting: in our next data sweep I will include brief measures of somatic disorders, including headaches and sleep problems.

We have measured resilience in nurses, and have found that some nurses can cope with much stress without suffering psychological burnout etc; but others do not, and are likely to leave the profession.

R.: We are very pleased that our study was reviewed by a person with a deep understanding of the risk of violence. In the past we have had the opportunity to conduct longitudinal studies on two different populations and these studies, which we have cited in this paper, have allowed us to demonstrate that violence induces an increase in perceived stress and that those who are in a job strain state. they are more exposed to violence than other workers. In another longitudinal study we are currently conducting in Covid-19 center staff, stress is associated with burnout and the intention to leave work. In this cross-sectional study, we did not measure burnout or the desire to leave work. What the reviewer gave us is excellent advice for a continuation of the study.

I would like the authors of this paper to add  to their discussion section (perhaps from line 338 onwards) a discussion, with additional references, of a variety of stressful factors faced by nurses, including the stresses of COVID care and associated patient mortality; and staff shortages (certainly in the  UK) because of these stressors, which may lead to downward spirals in patient care (e.g. long waits by patients for treatment and care leads to their aggressive behaviours). The authors must decide whether the somatic problems they measure may be related to more general malaise in nurses, and might predict burnout and intention to leave nursing.

R.: We sincerely thank the reviewer who gave us the opportunity to better frame this study in light of other previous experiences in which violence and stress were predictors of burnout and cessation of work. In response to the reviewer's stimulating remark, we have introduced the following paragraph in the article:

In the acknowledgments, the (excellent) work of their English-language advisor is noted with the name: Ms hank E.A. Wright. Is this a mistake - i.e. "hank" is a male name?

R.: We apologize for the typo and thank the reviewer for noticing. We corrected the name Elisabeth Ann Wright.

Reviewer 3 Report

Because I was interested in nurse bullying, I was happy to review this study. The purpose and results of the study were presented inconsistently, and some correction was required.

If the title is 'The impact of workplace violence on headache and sleep problems in nurses', why is an nurse assistants included in the participants? Please consider renaming the title to include the participants.

Please indicate the reliability of the HIT-6, PSQI, and ERI tools in this study.

Was the purpose of the study to compare the differences between nurses and nursing assistants? Please present the results according to the research questions and purposes.

Rather than writing statistical results in the result description section, it would be better to present them directly in the Table 1.

Why did we see correlation with age in bivariate correlation?

Adjusting the confounding variables in regression is considered an appropriate analysis.

Author Response

Because I was interested in nurse bullying, I was happy to review this study. The purpose and results of the study were presented inconsistently, and some correction was required.

Response: We thank the reviewer for the attention and care with which he / she has evaluated our study. As we explained in the Introduction, the aim was to evaluate the association between violence and neuropsychic disorders. We investigated a convenience sample, made up of the nursing staff of the healthcare company where we are responsible for health surveillance.

If the title is 'The impact of workplace violence on headache and sleep problems in nurses', why is an nurse assistants included in the participants? Please consider renaming the title to include the participants.

R.: As the reviewer rightly noted, we analyzed the relationship between violence and psychoneurological problems in all the nurses in the health company, whether they had a degree (registered nurses), or if they performed the simpler tasks that derive from professional training (assistant nurses). In the title the generic term of "nurse" includes the two categories, while in the text they are clearly differentiated.

Please indicate the reliability of the HIT-6, PSQI, and ERI tools in this study.

R.: We gladly agreed to the correct request of the reviewer, repeating the Cronbach alpha coefficients of the questionnaires.

Was the purpose of the study to compare the differences between nurses and nursing assistants? Please present the results according to the research questions and purposes.

R.: The aim of the study was not to compare the two categories of nurses with different qualifications, but to evaluate the relationship between violence and psycho-neurological symptoms in all those who perform nursing tasks. We made readers aware of a difference we found in reporting of non-physical violence, and we wondered if this difference corresponds to a real lower frequency of verbal aggression towards less qualified personnel, or if it is not produced by a lower propensity to these workers to report incivility. Only subsequent studies can solve this question.

Rather than writing statistical results in the result description section, it would be better to present them directly in the Table 1.

R.: We agree with the reviewer, and in fact we have included in table 1 all the results that can be contained in the table without compromising its readability and clarity. In the following text we have briefly commented on the table, trying to avoid repetition.

Why did we see correlation with age in bivariate correlation?

R.: We studied the correlation between all the continuous variables of the study; three of these (stress, headache impact, and sleep quality) were clinical, one (age) chronological. The correlation indicates that age worsens the quality of sleep but does not change the impact of the headache. We briefly commented on this result in the manuscript.

Adjusting the confounding variables in regression is considered an appropriate analysis.

R.: In this study, by means of logistic regression, we first assessed the relationship between different types of violence and neuropsychic disorders, then we adjusted for age, sex and professional category. Finally, we introduced stress for the known relationship between this variable and psychoneurological disorders. We thank the reviewer for approving the method.

Round 2

Reviewer 1 Report

Thanks for the reply. I encourage you to continue delving into the topic discussed